# Variation in Susceptibility to Downy Mildew Infection in Spanish Minority Vine Varieties

**DOI:** 10.3390/plants12142638

**Published:** 2023-07-13

**Authors:** Susana Boso, Pilar Gago, José-Luis Santiago, Gregorio Muñoz-Organero, Félix Cabello, Belén Puertas, Anna Puig, Carme Domingo, M. Esperanza Valdés, Daniel Moreno, Emilia Diaz-Losada, José F. Cibriain, Oier Dañobeitia-Artabe, José-Antonio Rubio-Cano, Jesús Martínez-Gascueña, Adela Mena-Morales, Camilo Chirivella, Jesús-Juan Usón, María-Carmen Martínez

**Affiliations:** 1Misión Biológica de Galicia, Consejo Superior de Investigaciones Científicas (CSIC), Carballeira 8, 36143 Salcedo, Spain; susanab@mbg.csic.es (S.B.); pgago@mbg.csic.es (P.G.); santi@mbg.csic.es (J.-L.S.); 2Instituto Madrileño de Investigación y Desarrollo Rural, Agrario y Alimentario (IMIDRA), Finca El Encín, Ctra. A-2 km 38, 28805 Alcalá de Henares, Spain; gregorio.munoz@madrid.org (G.M.-O.); felix.cabello@madrid.org (F.C.); 3Instituto de Investigación y Formación Agraria, Pesquera y de la Producción Ecológica, Ctra. Cañada de la Loba (CA-3101) Pk. 3.1, 11471 Jerez de la Frontera, Spain; mariab.puertas@juntadeandalucia.es; 4Catalan Institute of Vine and Wine—Institute of Agrifood Research and Technology (INCAVI-IRTA), Plaça Àgora 2, 08720 Vilafranca del Penedès, Spain; apuigpujol@gencat.cat (A.P.); carme.domingo@gencat.cat (C.D.); 5Center for Scientific and Technological Research of Extremadura (CICYTEX), Food and Agriculture, Technology Institute of Extremadura (INTAEX), Avenue Adolfo Suárez s/n, 06071 Badajoz, Spain; esperanza.valdes@juntaex.es (M.E.V.); daniel.moreno@juntaex.es (D.M.); 6Estación de Viticultura y Enología de Galicia (EVEGA), Ponte San Clodio s/n, 32419 Leiro, Spain; emilia.diaz.losada@xunta.gal; 7Estación de Viticultura y Enología de Navarra (EVENA), C. del Valle de Orba, 34, 31390 Olite, Spain; jf.cibriain.sabalza@navarra.es; 8Diputación Foral de Bizkaya, Avda. Lehendakari Aguirre, 9, 48014 Bilbao, Spain; oier.danobeitia@bizkaia.eus; 9Instituto Tecnológico Agrario de Castilla y León (ITACYL), Finca Zamadueñas, Ctra. Burgos km. 119, 47071 Valladolid, Spain; rubcanjo@itacyl.es; 10Instituto Regional de Investigación y Desarrollo Agroalimentario y Forestal de Castilla-La Mancha (IRIAF), Ctra. Toledo-Albacete s/n, 13700 Tomelloso, Spain; jmartinezg@jccm.es (J.M.-G.); amenam@jccm.es (A.M.-M.); 11Servicio de Producción Ecológica e Innovación, Instituto Tecnológico de Viticultura y Enología, Av. del General Pereyra, 4, 46340 Requena, Spain; chirivella_cam@gva.es; 12Centro Transferencia Agroalimentaria de Aragón, Avda. de Movera S/N, 50001 Zaragoza, Spain; jjuson@aragon.es

**Keywords:** fungal diseases, incidence, *Vitis vinifera*, downy mildew, minority varieties

## Abstract

Downy mildew is one of the most destructive diseases affecting grapevines (*Vitis vinifera* L.). Caused by the oomycete *Plasmopara viticola* (Berk. and Curt.) Berl. and de Toni, it can appear anywhere where vines are cultivated. It is habitually controlled by the application of phytosanitary agents (copper-based or systemic) at different stages of the vine growth cycle. This, however, is costly, can lead to reduced yields, has a considerable environmental impact, and its overuse close to harvest can cause fermentation problems. All grapevines are susceptible to this disease, although the degree of susceptibility differs between varieties. Market demands and European legislation on viticulture and the use of phytosanitary agents (art. 14 of Directive 128/2009/EC) now make it important to know the sensitivity of all available varieties, including minority varieties. Such knowledge allows for a more appropriate use of phytosanitary agents, fosters the commercial use of these varieties and thus increases the offer of wines associated with different *terroirs*, and helps identify material for use in crop improvement programmes via crossing or genetic transformation, etc. Over 2020–2021, the susceptibility to *P. viticola* of 63 minority vine varieties from different regions of Spain was examined in the laboratory using the leaf disc technique. Some 87% of these varieties were highly susceptible and 11% moderately susceptible; just 2% showed low susceptibility. The least susceptible of all was the variety Morate (Madrid, IMIDRA). Those showing intermediate susceptibility included the varieties Sanguina (Castilla la Mancha, IVICAM), Planta Mula (Comunidad Valenciana, ITVE), Rayada Melonera (Madrid, IMIDRA), Zamarrica (Galicia, EVEGA), Cariñena Roja (Cataluña, INCAVI), Mandrègue (Aragón, DGA) and Bastardo Blanco (Extremadura, CICYTEX). The highly susceptible varieties could be differentiated into three subgroups depending on sporulation severity and density.

## 1. Introduction

Downy mildew is, on a global level, one of the diseases that most affects grapevines (*Vitis vinifera* L.) [1]. It is caused by the oomycete *Plasmopara viticola* (Berk. and Curt.) Berl. and de Toni. Its control largely relies on the use of phytosanitary agents. Copper-based contact-acting products or systemic agents are commonly applied at different times during the vine growth cycle. Indeed, successful modern viticulture is dependent on the repeated application of large quantities of fungicides. Their use in Europe is particularly heavy. Not only does this have a negative environmental impact, it reduces the profitability of viticulture compared to the raising of other crops, calling into question the sustainability of wine production (a problem for both viticulturalists and consumers). The continued (and excessive) use of fungicides could also provoke the development of resistance to them, and the appearance of more aggressive, more virulent ‘races’ of the causal pathogen.

As far as we know, all grapevine varieties are susceptible to downy mildew, although not all to the same degree [2]; differences in susceptibility may also exist between clones of the same variety [3]. It is not known for sure how many varieties exist, but a figure somewhere between 8000 and 10,000 seems likely [4]. Despite the many planting options this affords, many winemaking regions around the world use only 10–12 varieties. Over the last 20 years, however, market competition has led to increased interest in the recovery of old, ‘pre-phylloxera’ varieties which, for a variety of reasons, fell into disuse (with some even approaching extinction) [5,6]. Some of these varieties are unknown beyond their local areas, yet they may have great oenological potential, and may even be able to adapt to different climates [5,6,7,8,9,10,11,12,13]. Certainly, our poor knowledge of the characteristics of these varieties extends to their degree of susceptibility to diseases such as downy mildew. Their study might allow some to come into commercial use, diversifying the offer of wines linked to different *terroirs*. Material that could be used in crop improvement programmes (crossing, genetic transformation, etc.) designed to produce plants resistant to disease, might also be identified. Such programmes, however, have so far only involved wild American (*V. riparia*, *V. rupestris*, *V. rotundifolia* and *V. cinerea*, etc.) and Far Eastern (*V. piasezkii*, *V. amurensis, V. romanetii*; *V. vinifera* Kishmish vatkana) species. These can confer partial or even total resistance to downy mildew (such is the case of *V. rotundifolia* and *V. piasezkii*) [14,15]. To date, 27 genomic regions associated with resistance to the disease (Rpv loci) have been identified [16,17,18,19,20]. Some of the genes that confer this resistance include Rpv1 and Rpv2 in *Muscadinia rotundifolia* Michaux, Rpv3 and Rpv19 in *Vitis rupestris* Scheele, Rpv4, Rpv7, Rpv11, Rpv17, Rpv18, Rpv20 and Rpv21 in unspecified American species, and Rpv5, Rpv6, Rpv9 and Rpv13 in *V. riparia*. Recently, three loci—*Rpv29*, *Rpv30* and *Rpv31*—have been identified in *V. vinifera* (Georgian germplasm) that confer resistance.

The aim of the present work was to determine the susceptibility to downy mildew of minority, pre-phylloxera vine varieties from different regions of Spain.

## 2. Materials and Methods

### 2.1. Plant Materials

Sixty-three minority varieties from 13 regions of Spain (Figure 1) were studied over 2020 and 2021. In January/February of each year, research groups in each of these regions sent 10–20 cuttings of varieties of interest to the MBG-CSIC for analysis. All cuttings were disinfected, placed in paraffin wax, and held in a cold chamber for four months to promote later root growth. Two rounds of bud break (one in March and one in April) were organised for each variety in each year, thus ensuring sufficient material for testing. For each round, cuttings were placed in water for a few hours in order to hydrate. A hormone-based rooting solution (0.4% indole butyric acid) was then applied to the base of the cuttings, and five of each variety that developed 3–4 buds were planted in a peat-perlite mixture in alveolar trays within a cultivation chamber (hot bed temperature 30 °C, air temperature 22 °C, humidity 80%, 8 h dark/16 h light). When rooting had taken place, the plants were transplanted to pots and placed in a greenhouse with a controlled environment (temperature 25 °C, humidity 80%, 8 h dark/16 h light). Materials belonging to the resistant rootstock varieties 110-Richter (derived from the cross *V. berlandieri cv. Rességuier n°2* × *V. rupestris cv. Martin*) and SO_4_ (*V. berlandieri* × *V. riparia; selection Oppenheim de Teléki n°4*) were planted to provide controls.

### 2.2. Collection and Preservation of Plasmopara viticola

*Plasmopara viticola* for inoculations was obtained from natural infections of vines at the MBG-CSIC research vineyard, following the method of Rumbolz et al. [21].

### 2.3. Inoculation of Leaf Discs

The leaf disc technique described by Rumbolz et al. [21] and Staudt et al. [22] was used to determine the susceptibility of the different varieties to downy mildew. Once the plants had produced canes some 15–30 cm in length, five leaves (the 5th or 6th leaves from the apical bud) were collected for each test and control variety. Thirty discs were then punched from the leaves and placed in Petri dishes (one variety per dish). Each disc was inoculated with *P. viticola* using a 50 μL of suspension of sporangia (50,000 mL^−1^) and left to incubate for six days (25 °C, 95% relative humidity, 8 h dark/16 h light). On the 6th day, each disc was visually inspected for signs of infection, measuring sporulation incidence (percentage of discs showing sporulation), sporulation severity (surface area affected by sporulation), and sporulation density (concentration of sporangia). The latter two variables were quantified using a visual scale (for both variables: 0–25% low; 25–50% moderate; 50–75% high; 75–100%: very high). Inoculations were performed twice, once in June and once in July, using the material produced in the two rounds of bud break (see above).

### 2.4. Statistical Analysis

Differences in the recorded variables between the varieties were analysed by two-way ANOVA (significance was recorded at *p* < 0.05, 0.01 and 0.001). The F test was then used to compare each fixed factor against its error. The means of those variables that returned a significant F value were then subjected to Fisher’s least significant difference (LSD) test. Principal components analysis was used to confirm the existence of groups of varieties with different levels of susceptibility. All analyses were made using SAS System v.8.1 software [23].

## 3. Results and Discussion

No differences were seen between the results for the plants produced in the first and second rounds of sprouting; the results for both were therefore analysed together. Table 1 shows, for each variety, the mean, standard deviation and coefficient of variation for each of the three measured variables. ANOVA revealed significant differences (*p* < 0.01) among the varieties and years for all three variables. The interaction *variety* × *year* had no significant impact on the variables recorded; thus, the plants of each variety behaved similarly in both years. The control varieties (110-Richter and SO_4_) showed resistance to the disease, returning sporulation incidence values of <15% and sporulation density values of <20%, confirming what was reported by other authors [2,24,25,26,27], with SO_4_ being the more resistant of the two (Table 1). The SO_4_ leaf discs showed small, dispersed necrotic spots, while those for 110-Richter showed fewer but larger sports. This necrosis is a response to infection seen in resistant vines [28,29,30]; hypersensitivity reactions cause programmed cell death around infection sites, helping to prevent the further spread of the pathogen.

Fisher’s LSD test revealed 87% of the test varieties to be highly susceptible to downy mildew, while 11% were moderately susceptible and just 2% showed low susceptibility (Figure 2). Some 87.5% showed a sporulation incidence of >75%, 10.9% returned values of 50–75%. The variety Morate (from the IMIDRA), however, returned an incidence value of <50% (34%). The variety Hebén from Extremadura returned the highest severity score (75%); overall, 29.68% of the varieties returned values of 50–75%, 14.6% returned values of 50–25%, and four of <15% (Tinto Jeromo, IVICAM; Mandrègue, Aragón-DGA; Albariño Tinto, Galicia-EVEGA; Planta Mula, Comunidad Valenciana-ITVE). Additionally, 11.1% of the test varieties showed a sporulation density of >75%, 67.2% returned a value of 50–75%, and the remainder (21.9%) a value of 25–50%.

These results show the great majority of the test varieties to be highly susceptible to downy mildew. The incidence of sporulation, however, was shown to be a poor discriminator for assessing degrees of susceptibility; sporulation severity and density were much better indicators.

Principal components analysis involving all three measured variables showed the first two principle components to explain 91% of the total variance. In the first component (Prin 1), sporulation severity and density had the greatest weight, while in the second component (Prin 2) the incidence of sporulation had the greatest weight (Figure 3). With respect to Prin 1, Trobat Negre (Cataluña, INCAVI), Hebén (Extremadura, CICYTEX), Zurieles (Castilla la Mancha, IVICAM) and Rayada Melonera (Andalucía, IFAPA) group towards the right of the graph given the high severity and density values they returned. Morate (Madrid, IMIDRA), which had the lowest severity and density results, was placed to the extreme left. With respect to Prin 2, Albariño Tinto (Galicia, EVEGA) appears in the upper part of the graph given its high incidence value; Morate (Madrid, IMIDRA), Trobat Negre (Cataluña, INCAVI), and Rayada Melonera (Andalucía, IFAPA) and the rootstock varieties 110-Richter and SO_4_ group towards the bottom given the lower incidence values they returned. Thus, three groups of varieties are distinguishable. The first is formed by Morate (Madrid, IMIDRA)—the least susceptible, with sporulation incidence, severity and density values all <40%. The second comprises the varieties showing moderate susceptibility (incidence < 75%, severity < 50%, density < 50%); within this latter group, the varieties Sanguina (Castilla la Mancha, IVICAM), Planta Mula (Comunidad Valenciana, ITVE), Rayada Melonera (Madrid, IMIDRA), Zamarrica (Galicia, EVEGA), Cariñena Roja (Cataluña, INCAVI), Mandrègue (Aragón, DGA) and Bastardo Blanco (Extremadura, CICYTEX) were the most resistant. Note that the last variety clustered with this group despite returning a sporulation density of 59% (Figure 4). Finally, the third group is formed by highly susceptible varieties. Within this group, three subgroups can be distinguished (Figure 5).

Subgroup A: varieties with an incidence value of >75% plus severity and density values of >50%, thus including Riera 2, Albana and Trobat Negre (Cataluña, INCAVI), Rufete Serrano, Hebén and Zurieles (Extremadura, CICYTEX), Albillo del Pozo, Zurieles and Terriza (Castilla La Mancha, IVICAM), Diega 1 and Tortozona Tinta (CF Navarra, EVENA + UPNA), Tinto Jeromo and Gajo Arroba (Castilla y León, ITACYL), Jarrosuelto and Tortozona Tinta (Aragón, DGA), Rayada Melonera and Indiana (Andalucía, IFAPA), and Forcallat and Planta Nova (Comunidad Valenciana, ITVE).

*Subgroup B*: varieties with an incidence value of >75%, plus a severity value of <50%, but a density value of >50%, thus including Sanguina and Riera 46 (Cataluña, INCAVI), Santa Fe, Castellana Blanca, Jarrosuelto and Diega 2 (CF Navarra, EVENA + UPNA), Montonera del Casar, Jarrosuelto and Maquías (Castilla La Mancha, IVICAM), Albarín Tinto and Xafardán (Galicia, EVEGA), Tortozona Tinta and Cagarrizo (Madrid, IMIDRA), Greta and Santa Fe (Aragón, DGA), Rufete Serrano and Estaladiña (Castilla y León, ITACYIL), Corchera (Andalucía, IFAPA), Arcos (Comunidad Valenciana, ITVE), and Giro Negre (Baleares, IUB).

*Subgroup C*: varieties with an incidence value of >75%, plus severity and density values of <50%, thus including Moscatel de Grano Menudo, Castellana Blanca, Tinto Fragoso, Tortozona Tinta and Tinto Jeromo (Castilla la Mancha, IVICAM), Ratiño, Albariño Tinto and Albillo do Avia (Galicia, EVEGA), Cenicienta (Castilla y León, ITACYL), Evena 1 (CF Navarra, EVENA + UPNA), Cagarrizo (Extremadura, CICYTEX), Albana (Aragón, DGA), Terriza, Hebén and Castellana Blanca (Madrid, IMIDRA), and Riera 43 (Cataluña, INCAVI).

The present results thus show that the majority of the tested varieties were either highly or moderately susceptible to downy mildew, although it should be noted that the severity of sporulation (i.e., the surface area occupied by sporulating oomycete) was often <50% (Figure 6). Thus, once the pathogen has entered the plant, response mechanisms would seem to come into play that are able to defend the host to a degree depending on the variety. Similar results were reported by Hernández et al. [31] for some of the same varieties when challenged with the same pathogen.

Plants of the variety Rayada Melonera from the IFAPA were more susceptible (in terms of sporulation incidence, severity and density) compared to those from the IMIDRA. Similarly, Hebén material from Extremadura was more susceptible than the same from the IMIDRA, and significant differences were seen in terms of incidence between Castellana Blanca material from the IVICAM and both the IMIDRA and EVENA. In a further difference between these latter materials, the plants from the EVENA returned significantly higher sporulation density values. Other authors [32] have reported similar behaviour within the varieties Chasselas Doré and Doña Blanca, with resistances to downy mildew and powdery mildew (caused by *Erysiphe necator*) varying between low and moderate depending on the material’s origin. This phenomenon might be explained by clonal differences, agronomic- or climate-linked factors, etc. Some authors attribute host–pathogen interactions to the adaptation of plants to the damage caused to the former by the latter [33]. Thus, plants of the same variety but of different origin, and perhaps therefore living in different environments or exposed to pathogen strains of different aggressiveness, may develop different levels of resistance to further attack.

The clustering of the present varieties in terms of their susceptibility was independent of berry colour. It might be thought that red varieties, with their higher contents in phenolic compounds such as resveratrol [34,35], might be less susceptible to downy mildew [28,36]. Certainly this has been reported with respect to other pathogens such as *Botrytis cinerea* [37,38]. However, no such correlation was seen in the present work, nor in previous work was any detected in different vine varieties growing in the field [2].

Colleagues working in tandem within the same major project (RTI 2018-101085-RC32—see Acknowledgements) detected no relationship between susceptibility to *P. viticola* and agronomic or ampelographic factors such as earliness or the density of reclining or erect hairs (which provide a barrier to zoospores trying to enter the plant via the stomata) (at press). Other reports [39,40] also indicate a lack of relationship between susceptibility to downy mildew and leaf morphology variables such as hair density. However, these authors indicated associations between factors such as stomatal density and the production of stilbenes; these factors would appear to influence the progress of disease.

## 4. Conclusions

Importantly, 2% of the tested varieties—in particular Morate—showed low susceptibility to the pathogen. If cultivated, they might require fewer fungicide treatments than other varieties, reducing costs as well as the environmental impact associated with the use of these agents. They might also be good material for the production of resistant varieties. Their cultivation would not infringe current viticultural legislation, and might improve the selection of wines available to consumers.

## Figures and Tables

**Figure 1 plants-12-02638-f001:**
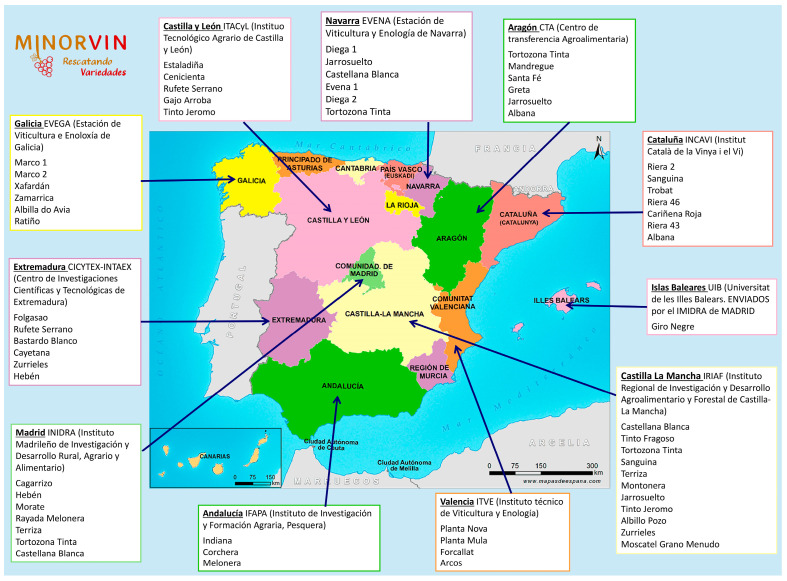
Provenance of the cuttings used in the present work.

**Figure 2 plants-12-02638-f002:**
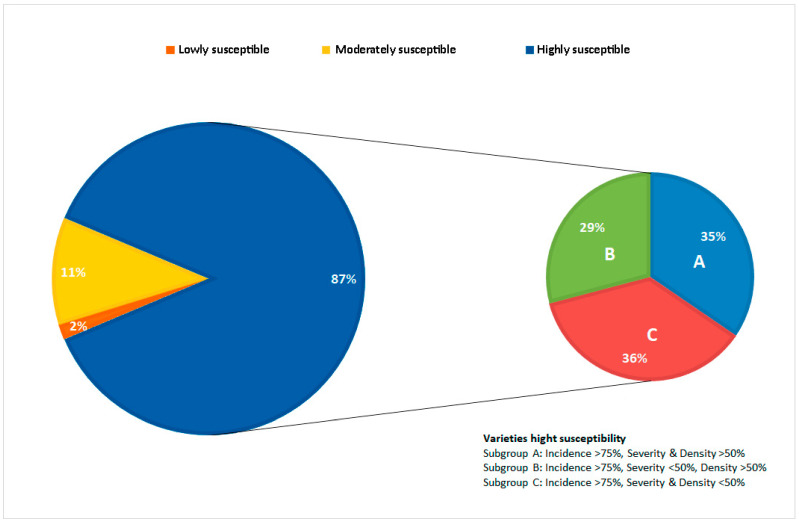
Percentage of susceptibility in the most susceptible varieties, taking into account disease incidence, severity and density.

**Figure 3 plants-12-02638-f003:**
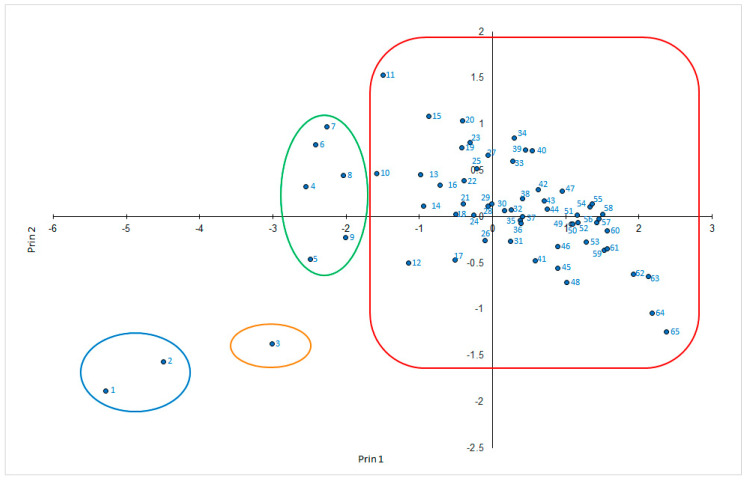
Principal components analysis involving the three variables recorded: 1—SO_4_; 2—110-Richter; 3—Morate (MIDRA); 4—Zamarrica (EVEGA); 5—Mandrègue (DGA); 6—Planta Mula (ITVE); 7—Tinto Jeromo (IVICAM); 8—Sanguina (IVICAM); 9—Cariñena Roja (INCAVI); 10—Castellana Blanca (IMIDRA); 11—Albariño Tinto (EVEGA); 12—Rayada Melonera (IMIDRA); 13—Hebén (IMIDRA); 14—Riera 43 (INCAVI); 15—Tinto Fragoso (IVICAM); 16—Albana (DGA); 17—Bastardo Blanco (CICYTEX); 18—Terriza (IMIDRA); 19—Cagarrizo/Folgasao (CICYTEX); 20—Castellana Blanca (IVICAM); 21—Maquías (IVICAM); 22—Arcos (ITVE); 23—Cenicienta (ITACYL); 24—Tortozona Tinta (IVICAM); 25—Albilla do Avia (EVEGA); 26—Xafardán (EVEGA); 27—Santa Fe/Cadrete (EVENA + UPNA); 28—Cagarrizo (IMIDRA); 29—Estaladiña (ITACYL); 30—Riera 46 (INCAVI); 31—Sanguina (INCAVI); 32—Diega 2 (EVENA + UPNA); 33—Evena1 (EVENA + UPNA); 34—Moscatel Grano Menudo (IVICAM); 35—Rufete Serrano (ITACYL); 36—Albarin Tinto (EVEGA); 37—Castellana Blanca (EVENA + UPNA); 38—Gajo Arroba (ITACYL); 39—Ratiño (EVEGA); 40—Greta (DGA); 41—Planta Nova (ITVE); 42—Tortozona Tinta (IMIDRA); 43—Montonera del Casar (IVICAM); 44—Corchera (IFAPA); 45—Jarrosuleto (EVENA + UPNA); 46—Jarrosuleto (IVICAM); 47—Santa Fe (DGA); 48—Terriza (IVICAM); 49—Tortozona Tinta (DGA); 50—Albillo del Pozo (IVICAM); 51—Tinto Jeromo (ITACYL); 52—Albana (INCAVI); 53—Tortozona Tinta (EVENA + UPNA); 54—Giro Negre (UIB); 55—Rufete Serrano (CICYTEX); 56—Zurieles (CICYTEX); 57—Diega 1 (EVENA + UPNA); 58—Riera 2 (INCAVI); 59—Forcallat (ITVE); 60—Indiana (IFAPA); 61—Jarrosuelto (DGA); 62—Zurieles (IVICAM); 63—Hebén (CICYTEX); 64—Trobat Negre (INCAVI); 65—Rayada Melonera (IFAPA).

**Figure 4 plants-12-02638-f004:**
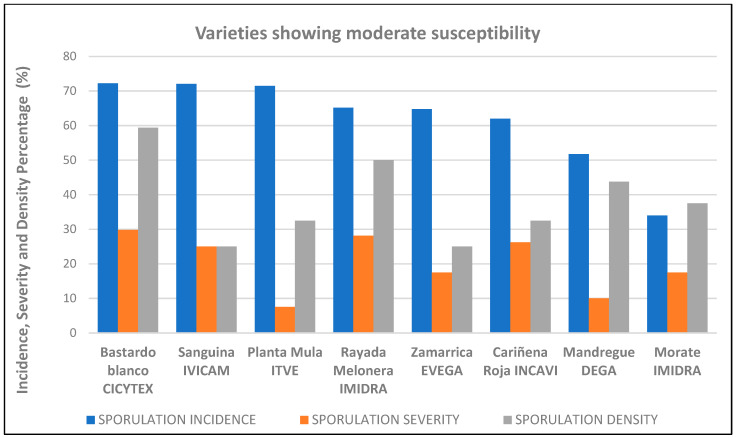
Percentage susceptibility in the ‘intermediate susceptiblility’ varieties, taking into account disease incidence, severity and density.

**Figure 5 plants-12-02638-f005:**
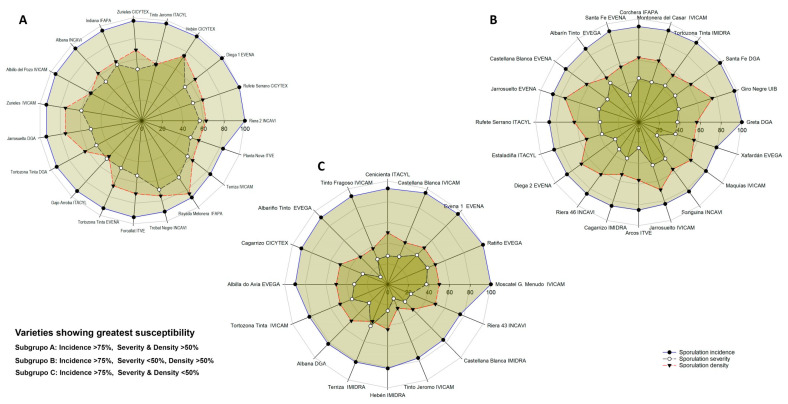
Subgroups of the susceptible varieties. (**A**): incidence > 75%, severity and density > 50%; (**B**): incidence > 75%, severity < 50%, density > 50%; (**C**): incidence > 75%, severity and density < 50%.

**Figure 6 plants-12-02638-f006:**
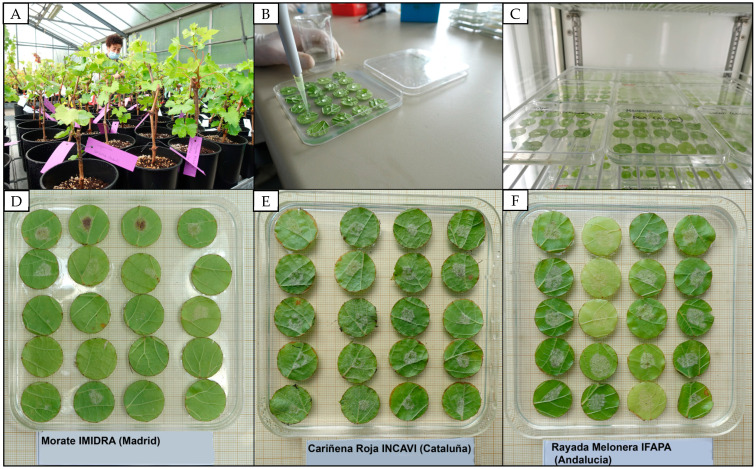
Images of leaf discs showing different degrees of resistance to downy mildew (6 dpi). (**A**) Plants grown in the greenhouse for leaf collection. (**B**) Inoculation of the discs. (**C**) Incubation of the discs in an environmental chamber. (**D**) Least susceptible variety. (**E**) Example of moderate susceptibility. (**F**) Most susceptible variety.

**Table 1 plants-12-02638-t001:** Sporulation incidence, severity and density with respect to the different vine varieties tested. Values are means. SD = standard deviation, CV = coefficient of variation, LSD = least significant difference.

		Incidence			Severity (%)			Density (%)		
Variety	Origin	Mean	SD	CV	Mean	SD	CV	Mean	SD	CV
Greta	DGA	100.0 a	0.0	0.0	37.5 efghijkl	17.7	47.1	56.2 bcdefg	23.9	42.5
Moscatel G. Menudo	IVICAM	100.0 a	0.0	0.0	37.5 efghijkl	14.4	38.5	50.0 bcdefgh	28.9	57.7
Ratiño	EVEGA	100.0 a	0.0	0.0	41.7 defghijk	14.4	34.6	50.0 bcdefgh	0.0	0.0
Riera 2	INCAVI	100.0 a	0.0	0.0	56.2 abcdef	12.5	22.2	62.5 abcde	25.0	40.0
Rufete Serrano	CICYTEX	100.0 a	0.0	0.0	52.5 bcdefg	16.6	31.6	62.5 abcde	14.4	23.1
Diega 1	EVENA + UPNA	98.4 ab	3.1	3.2	53.1 abcdefg	15.7	29.6	65.6 abcd	12.0	18.2
Giro Negre	UIB	97.5 ab	5.0	5.1	40.6 defghijk	18.7	46.1	75.0 ab	0.0	0.0
Hebén	CICYTEX	97.5 ab	17.8	19.9	75.0 a	23.9	34.8	75.0 ab	12.5	18.2
Santa Fe	DGA	97.5 ab	5.0	5.1	43.7 cdefghijk	23.9	54.7	62.5 abcde	25.0	40.0
Tinto Jeromo	ITACYL	97.0 ab	3.6	3.7	56.5 abcdef	16.1	28.7	56.2 bcdefg	23.9	42.5
Zurieles	CICYTEX	97.0 ab	6.0	6.2	50.0 bcdefghi	0.00	0.0	68.7 abcd	12.5	18.2
Evena 1	EVENA + UPNA	96.2 ab	7.5	7.8	40.0 efghijk	12.2	30.6	50.0 bcdefgh	20.4	40.8
Castellana Blanca	IVICAM	95.7 ab	8.6	8.9	29.0 ijklmn	8.9	30.7	43.7 dfgh	12.5	28.5
Indiana	IFAPA	95.2 ab	3.5	3.6	59.4 abcde	12.0	20.2	62.5 abcde	14.4	23.1
Albana	INCAVI	95.0 abc	9.9	10.4	51.5 bcdefgh	22.1	42.6	62.5 abcde	14.4	23.1
Albillo del Pozo	IVICAM	95.0 abcd	5.77	6.08	56.2 abcdef	23.9	42.5	56.2 bcdefg	12.5	22.2
Tortozona Tinta	IMIDRA	95.0 abcd	10.0	10.5	43.7 cdefghijk	12.5	28.6	56.2 bcdefg	12.5	22.2
Jarrosuelto	DGA	93.7 abcde	12.5	13.3	50.0 bcdefghi	20.4	40.8	75.0 ab	0.0	0.0
Tortozona Tinta	DGA	93.7 abcde	12.5	13.3	50.0 bcdefghi	10.2	20.4	62.5 abcde	14.4	23.1
Zurieles	IVICAM	93.7 abcde	12.5	13.3	59.4 abcde	23.7	39.8	75.0 ab	20.4	27.2
Forcallat	ITVE	93.6 abcde	11.0	11.7	53.3 abcdefg	20.2	37.89	70.8 abc	7.2	10.2
Montonera del Casar	IVICAM	93.2 abcde	9.4	10.1	40.6 defghijk	23.7	58.2	62.5 abcde	14.4	23.1
Cenicienta	ITACYL	92.7 abcde	9.1	9.9	27.5 ijklmn	26.0	94.5	50.0 bcdefgh	20.4	40.8
Corchera	IFAPA	92.5 abcde	15.0	16.2	42.5 defghijk	12.0	28.0	62.5 abcde	17.7	28.3
Gajo Arroba	ITACYL	92.5 abcde	11.9	12.9	50.0 bcdefghi	20.4	40.8	46.9 cdefgh	15.7	33.5
Tortozona Tinta	EVENA + UPNA	92.5 abcde	11.9	12.9	50.0 ijklmn	20.4	40.8	68.7 bcdefg	37.5	54.5
Santa Fe/Cadrete	EVENA + UPNA	92.5 abcde	15.0	16.2	27.5 bcdefghi	5.0	18.1	56.2 abcd	23.9	42.5
Tinto Fragoso	IVICAM	92.2 abcde	11.8	12.8	26.2 jklmno	18.4	70.2	37.5 efghi	25.0	66.7
Albariño Tinto	EVEGA	91.5 abcde	13.9	15.2	10.0 no	10.0	100.0	37.5 efghi	25.0	66.7
Trobat Negre	INCAVI	90.8 abcde	12.7	14.0	68.7 ab	23.9	34.8	75.0 ab	0.0	0.0
Cagarrizo	CICYTEX	90.7 abcde	11.1	12.3	26.2 jklmno	18.4	70.2	50.0 bcdefgh	0.0	0.0
Albilla do Avia	EVEGA	89.8 abcde	16.7	18.6	32.5 ghijklm	21.8	67.1	50.0 bcdefgh	28.9	57.7
Rayada Melonera	IFAPA	88.8 abcde	22.5	25.3	65.6 abc	31.2	47.6	84.4 a	18.7	22.2
Albarín Tinto	EVEGA	87.7 abcdef	14.8	16.9	46.9 bcdefghij	15.7	33.5	53.1 bcdefg	21.3	40.2
Castellana Blanca	EVENA + UPNA	87.5 abcdef	14.4	16.5	37.5 efghijkl	14.4	38.5	62.5 abcde	14.4	23.1
Jarrosuelto	EVENA + UPNA	87.5 abcdef	14.4	16.5	40.6 defghijk	23.9	54.7	75.0 ab	12.5	18.2
Rufete Serrano	ITACYL	86.7 abcdef	18.9	21.7	37.5 efghijkl	14.4	38.5	62.5 abcde	14.4	23.1
Estaladiña	ITACYL	86.5 abcdef	27.5	31.9	37.5 efghijkl	14.4	38.5	53.1 bcdefg	21.3	40.2
Diega 2	EVENA + UPNA	86.0 abcdef	16.3	19.0	28.1 ijklmn	6.2	22.2	68.7 abcd	12.5	18.2
Riera 46	INCAVI	85.7 abcdef	16.7	19.5	32.5 ghijklm	21.8	67.1	62.5 abcde	14.4	23.1
Cagarrizo	IMIDRA	85.2 abcdefg	20.7	24.3	36.9 fghijklm	10.3	27.9	53.1 bcdefg	6.2	11.8
Arcos	ITVE	85.0 abcdefg	30.0	35.3	25.0 jklmno	56.6	29.8	56.2 bcdefg	22.2	59.4
Terriza	IVICAM	84.2 abcdefg	11.9	14.1	56.2 abcdef	12.5	22.2	62.5 abcde	14.4	23.1
Planta Nova	ITVE	83.3 abcdefg	28.9	34.6	50.0 bcdefghi	0.0	0.0	58.3 bcdef	28.8	49.5
Jarrosuelto	IVICAM	83.2 abcdefg	22.2	26.7	43.7 cdefghijk	12.0	29.5	68.7 abcd	0.0	0.0
Sanguina	INCAVI	83.0 abcdefg	19.9	24.0	43.7 cdefghijk	16.14	36.9	56.2 bcdefg	23.9	42.5
Tortozona Tinta	IVICAM	82.2 abcdefg	12.0	14.6	37.5 efghijkl	14.4	38.5	50.0 bcdefgh	20.4	40.8
Albana	DGA	81.7 abcdefg	21.3	26.1	25.6 jklmno	9.2	36.0	50.0 bcdefgh	20.4	40.8
Hebén	IMIDRA	81.2 abcdefg	13.5	16.2	25.6 jklmno	9.2	36.0	43.7 dfgh	12.5	28.6
Terriza	IMIDRA	81.2 abcdefg	23.9	29.5	43.7 cdefghijk	23.9	54.7	38.7 efghi	30.34	78.4
Maquías	IVICAM	80.6 abcdefg	4.1	5.1	21.9 klmno	10.3	47.0	62.5 abcde	14.4	23.1
Xafardán	EVEGA	79.2 abcdefg	24.1	30.4	37.5 efghijkl	25.0	66.7	56.2 bcdefg	37.5	66.7
Tinto Jeromo	IVICAM	77.0 abcdefg	28.0	36.4	15.0 mno	11.5	77.0	25.0 hi	0.0	0.0
Castellana Blanca	IMIDRA	76.0 bcdefg	28.3	37.1	23.8 klmno	16.4	69.0	34.4 fghi	12.0	34.8
Riera 43	INCAVI	76.0 bcdefg	7.4	9.8	24.4 klmno	22.9	94.1	50.0 bcdefgh	28.9	57.7
Sanguina	IVICAM	72.1 cdefgh	16.0	22.2	25.0 hijklmn	0.0	0.0	25.0 abcdef	0.0	0.0
Planta Mula	ITVE	71.5 defgh	34.6	48.4	7.5 jklmno	5.0	66.7	32.5 hi	29.9	91.9
Bastardo blanco	CICYTEX	72.2 efgh	26.9	37.7	29.8 no	18.8	63.3	59.4 ghi	12.0	20.2
Rayada Melonera	IMIDRA	65.2 fgh	11.7	17.9	28.1 ijklmn	6.2	22.2	50.0 bcdefgh	20.4	40.8
Zamarrica	EVEGA	64.5 fgh	23.1	35.7	17.5 lmno	15.0	85.7	25.0 hi	0.0	0.0
Cariñena Roja	INCAVI	62.0 hg	26.9	43.4	26.2 jklmno	25.5	97.2	32.5 ghi	29.9	91.9
Mandrègue	DGA	51.7 hi	14.20	27.4	10.0 no	10.0	100.0	43.7 dfgh	12.5	28.6
Morate	IMIDRA	34.0 ij	14.28	42.0	17.5 lmno	15.0	85.7	37.5 efghi	14.4	38.5
110-Richter	MBG	18.3 jk	11.25	62.1	15.0 mno	0.0	0.0	15.0 i	11.5	77.0
SO_4_	MBG	5.0 k	0.00	0.0	5.0 o	0.0	0.0	15.0 i	11.5	77.0
**LSD (0.05)**		23.5			22.4			25.4		

Values followed by the same letter, in each column, and for each variable, are not significantly different.

## Data Availability

The data presented in this study are available on request from the corresponding author.

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
