# Peer review of "Variation in Susceptibility to Downy Mildew Infection in Spanish Minority Vine Varieties"

_plants, 2023, doi:10.3390/plants12142638_

Round 1

Reviewer 1 Report

General comments

The manuscript describes the susceptibility of minor Spanish grapevine varieties to downy mildew. From 2020 to 2021, the laboratory examined the susceptibility to P. viticola of 63 minority grapevine varieties from different regions of Spain using the leaf disc technique. Two percent of the tested varieties showed low susceptibility to downy mildew, while 11% of the varieties were moderately susceptible. This study is of great importance to the grapevine community due to the increasing interest in resistant grapevine varieties. Additionally, this study describes the variation in sensitivity to downy mildew within Vitis vinifera minor varieties, which could regain interest for revitalization and use in breeding.

The manuscript is well-organized but simple. The English grammar needs improvement, and there are some parts that require revision. The introduction should provide more information justifying this study, including recently published information about grapevine resistance to downy mildew, phenotyping, and possible genetic backgrounds for resistance in grapevine. It is unclear whether a one-way or two-way ANOVA was performed, despite the author's statement in line 90: "The interaction variety x year had no significant impact on the variables recorded." The use of the term "interaction" implies a two-way ANOVA.

In my opinion, the manuscript has the potential for acceptance after major revision.

 Minor comments:

 Line 32: "grapevine" is usually written as one word, not two separate words like "grape vine." Please correct throughout the manuscript.

Lines 47-51: Is it necessary to include data in parentheses? For example, "variety Morate (Madrid, IMIDRA)." Origin of variety is important data but need to be written somewhere in the M&M section. Please correct throughout the manuscript.

Line 90: "among" varieties instead of "between."

Table 1a: Some values were written with one, two, or without decimal places. Please use one decimal place for all values.

Figure 2: The caption does not explain the presented results. Please clearly describe the presented results in the Figure 2 caption.

Lines 149-162: The numbers connected with varieties (written in italics) do not correspond to the text in Figure 3. This connection between numbers and varieties should be written in the Materials and Methods section or put in the caption of Figure 3.

Figure 4: The caption needs better explanation. Using just the percentage (%) is too general. Please provide the full variable name.

Figure 6: There are six different photos, but only four letters are used to describe the corresponding photos. Each photo illustration must be marked and explained in the caption.

Figure 1: Unfortunately, the resolution of the illustration is low. Could the authors provide an illustration with better resolution or sort the table with the full list of varieties used in the study and provide additional ampelographic information about the varieties (e.g., color of berry skin, main utility of grapes, wine or table)?

The English grammar needs improvements. 

Author Response

Reviewer 1

  • The English grammar needs improvement, and there are some parts that require revision.

Our use of English has been checked by an English science editor and writer (Adrian Burton: www.physicalevidence.es), with long experience in the preparation of scientific texts. We are quite sure it is of the standard required.

  • The introduction should provide more information justifying this study, including recently published information about grapevine resistance to downy mildew, phenotyping, and possible genetic backgrounds for resistance in grapevine.

More information has been provided in line with the reviewer's request.

  • It is unclear whether a one-way or two-way ANOVA was performed, despite the author's statement in line 90: "The interaction variety x year had no significant impact on the variables recorded.

It was a two-way ANOVA (now clarified in the text).  The fixed factor was 'varieties' and the random factor 'years' (interaction variety x year).

Line 32: "grapevine" is usually written as one word, not two separate words like "grape vine." Please correct throughout the manuscript.

Changed as requested.

Lines 47-51: Is it necessary to include data in parentheses?

We believe so since some varieties are from different places.  

Line 90: "among" varieties instead of "between."

Changed as requested.

Table 1a: Some values were written with one, two, or without decimal places. Please use one decimal place for all values.

Changed as requested.

Figure 2: The caption does not explain the presented results. Please clearly describe the presented results in the Figure 2 caption.

Clarification has been made.

Lines 149-162: The numbers connected with varieties (written in italics) do not correspond to the text in Figure 3. This connection between numbers and varieties should be written in the Materials and Methods section or put in the caption of Figure 3.

This has been corrected.

Figure 4: The caption needs better explanation. Using just the percentage (%) is too general. Please provide the full variable name.

Changed as requested.

Figure 6: There are six different photos, but only four letters are used to describe the corresponding photos. Each photo illustration must be marked and explained in the caption.

Corrected as suggested.

Figure 1: Unfortunately, the resolution of the illustration is low. Could the authors provide an illustration with better resolution or sort the table with the full list of varieties used in the study and provide additional ampelographic information about the varieties (e.g., colour of berry skin, main utility of grapes, wine or table)?

We now provide a higher resolution image.

Reviewer 2 Report

Dear Authors, I have read and reviewed the MS entitled "Variation in Susceptibility to Downy Mildew Infection in Spanish Minority Vine Varieties".

I have some observations:

-please check and correct all over the article the word Richter, when talking about grafting materials

-please check and correct according to journal requirements the order of the chapters. Is results and discussions before material and methods?

-why is figure 1 (towards the end of the article) after figure 6? Please correct the order in which info is presented in the MS

Please add info on how analysis regarding susceptibility to Plasmopara were made. Just visually?

I believe the MS needs more info, for example on genetic characteristics of each vine variety and its influence on fighting the disease.

The English language is mostly correct, there are some spelling issues that need to be corrected.

Author Response

Reviewer 2

  • Please check and correct all over the article the word Richter, when talking about grafting materials

This has been corrected.

  • Please check and correct according to journal requirements the order of the chapters. Is results and discussions before material and methods?

The order has been changed as suggested.

why is figure 1 (towards the end of the article) after figure 6? Please correct the order in which info is presented in the MS

This has been fixed.

Please add info on how analysis regarding susceptibility to Plasmopara were made. Just visually?

All assessments were made by the same person to avoid observer differences.

Round 2

Reviewer 1 Report

The authors have significantly improved the manuscript. The introduction now better justifies why downy mildew susceptibility in Vitis vinifera cultivars needs to be researched. The genetic sources of downy mildew resistance are mainly present in wild Vitis spp. although some genes are also present in V. vinifera cultivars and need to be studied in detail. Phenotyping of downy mildew susceptibility is certainly the most important step along the way.

The manuscript has been significantly improved by correcting other minor errors. The authors have removed all doubts that had arisen in the first version of the manuscript. I recommend accepting this work in this form after correcting few minor technical mistakes.

Minor suggestions:

Table 1 a and 1 b: round values to one decimal place (mean, SD and CV)

Figure 6: The letter must be in the position of the photo, but not abutting the caption text

Author Response

Rewiewer 1. Minor suggestions:

Table 1 a and 1 b: round values to one decimal place (mean, SD and CV)

This has been corrected.

Figure 6: The letter must be in the position of the photo, but not abutting the caption text

Changed as requested.

Reviewer 2 Report

Dear Authors, thank you for the revised version. I have read the  MS and I have some more minor observations:

-line 102, 132, what does sprouting mean? Were there seeds involved? Maybe you are reffering to bud break (the growth of buds and appearance of young leaves)? Please check and correct all over text

-line 111, please correct the name Richter

-line 114, in line with the figure 1, there is [SB1]. What is this?

-please check the journal requirements and correct if necessary the writing of degrees Celsius. Is the line under º correct? 

-please check and correct according to journal requirements all over the text the 2 spaces after a full stop.

-please check and correct according to journal requirements all over the text the formatting of the figure titles and the figures. Should it be centered?

-figure 4 - please reformat so writing is visible all over

-figure 6 - please reformat so that letters D, E, F do not cover the title of the figure

-please format according to journal requirements table 1 a and b so that values can be on one line. Maybe change from portrait to landscape?

-please format according to journal requirements table 1 a and b the letters indicating statistical significance. Is it not superscript?

-

Author Response

Rewiewer 2.  Minor suggestions:

-line 102, 132, what does sprouting mean? Were there seeds involved? Maybe you are reffering to bud break (the growth of buds and appearance of young leaves)? Please check and correct all over text

Clarification has been made.

-line 111, please correct the name Richter

This has been corrected.

-line 114, in line with the figure 1, there is [SB1]. What is this?

This has been eliminated

-please check the journal requirements and correct if necessary the writing of degrees Celsius. Is the line under º correct? 

Changed as requested.

-please check and correct according to journal requirements all over the text the 2 spaces after a full stop.

Changed as requested.

-please check and correct according to journal requirements all over the text the formatting of the figure titles and the figures. Should it be centered?

This has been corrected.

-figure 4 - please reformat so writing is visible all over

This has been corrected.

-figure 6 - please reformat so that letters D, E, F do not cover the title of the figure

This has been corrected.

-please format according to journal requirements table 1 a and b so that values can be on one line. Maybe change from portrait to landscape? -please format according to journal requirements table 1 a and b the letters indicating statistical significance. Is it not superscript?

We now provide a higher resolution of the table 1 according to journal of requirements